# Mechanisms of variability of decadal sea-level trends in the Baltic Sea over the 20th century

Sitar Karabil, Eduardo Zorita, Birgit Hünicke

Institute of Coastal Research, Helmholtz-Zentrum Geesthacht. Max-Planck Str.1, Geesthacht, 21502, Germany

*Correspondence to*: Sitar Karabil (starkarabil@gmail.com)

**Abstract.** Coastal sea-level trends in the Baltic Sea display decadal-scale variations around a long-term centennial trend. In this study, we analyse the spatial and temporal characteristics of the decadal trend variations and investigate the links between coastal sea-level trends and atmospheric forcing on decadal time scale. For this analysis, we use monthly means of sea level and climatic data sets. Sea level data set is composed of long tide gauge records and gridded sea-surface-height

(SSH) reconstructions. Climatic data sets are composed of sea level pressure, air temperature, precipitation, evaporation and climatic variability indices. The analysis indicates that atmospheric forcing is a driving factor of decadal sea-level trends. However, its effect is geographically heterogeneous.  This impact is large in the northern and eastern regions of the Baltic Sea.  In the southern Baltic Sea area, the impacts of atmospheric circulation on decadal sea level trends are smaller.

To identify the influence of the large-scale factors other than the effect of atmospheric circulation in the same season on the

Baltic Sea level trends, we filter out, for each season separately, the direct signature of atmospheric circulation on the Baltic Sea level by a multivariate linear regression model and analysed the residuals of this regression model. These residuals hint at a common underlying factor that coherently drives the decadal sea-level trends in the whole Baltic Sea. We found that this underlying effect is partly a consequence of decadal precipitation trends in the Baltic Sea basin in the previous season.

The investigation on the relation between the AMO-index and sea-level trends implies that this detected underlying factor is

not connected to oceanic forcing driven from the North Atlantic region.

**Keywords:** sea level, the Baltic Sea, decadal trend, statistical analysis.

## 1 Introduction

Global mean sea-level trend has risen over the 20th century with an approximate rate of 1.7 mm yr$^{-1}$, with higher rates of about 3.2 mm yr$^{-1}$ as measured by satellite altimetry over the past 30 years (Church and White, 2011; Nerem et al., 2010).

This rise is also projected to continue at high rates in the future due to the global warming. However, regional sea-level change has displayed and will likely continue to display clear deviations from the global sea-level average (Slangen et al., 2012; Church et al., 2013). As these authors pointed out, the sea-level trends estimated by satellite altimetry in the North Atlantic at high latitudes south of Greenland are of the order of 10 mm yr$^{-1}$, whereas immediately further south in the mid-

latitude North Atlantic, the sea-level trends may become negative, of the order of -5 mm yr$^{-1}$. In the Pacific Ocean, similar contrasts can be also observed. In the Tropical Western Pacific, these trends may attain a value of 15 mm yr$^{-1}$, whereas in the mid-latitude Eastern Pacific the trends are negative, of the order of -5 mm yr$^{-1}$. These regional differences in observed regional sea-level trends are likely caused by spatially heterogeneous atmospheric forcing and ocean internal variability (e.g.

Church et al., 2010). Since the regional trends can be sustained for several decades (Hu and Deser, 2013), the understanding of the origin of these deviations is important for more accurate predictions of regional sea-level rise (Carson et al. 2016; Cazenave and Llovel, 2010; Milne et al., 2009). This study contributes to the understanding of variations in the decadal sea-level trends in the Baltic Sea.

In the case of semi-enclosed seas like the Baltic Sea, the deviations of the sea-level trends from the global mean could

potentially be large since they are exposed to additional forcing, such as regional wind forcing and its interaction with the coastlines, the balance between precipitation and evaporation, and spill-over effects from the open ocean. Also, the regional oceanographic characteristics such as stratification due to regional temperature and salinity profiles may modulate the heat-uptake differently as in the open ocean.

Previous studies have shown that the sea-level records display relatively large variations of decadal trends in the Baltic Sea

(Richter et al., 2011). For instance, the Warnemünde (southern Baltic) sea-level record displays an average long-term sea-level trend over the last 150 years of about 1.5 mm yr$^{-1}$ (to a large extent caused by the Glacial Isostatic Adjustment that occurs at millennial timescales, as briefly explained later). However, the trends of the Warnemünde sea-level record calculated over 30-year windows may vary between -0.5 mm yr$^{-1}$ to 2.5 mm yr$^{-1}$. Although the recent 30-year trends are high, the maximum 30-year trend so far was reached around year 1900. This indicates that natural variations can cause

substantial deviations that should be understood and taken into account, especially for shorter term (multidecadal) future sea-level projections.

Whereas the mechanisms responsible for the interannual sea-level variability have been more profusely studied, it is still not known whether the mechanisms that have been found to account for the interannual variations of sea-level are also responsible for the variability of decadal sea-level trends in the Baltic Sea. It could be plausible that once the strong

interannual variations of the atmospheric circulation are filtered out, the decadal sea-level trends may be still affected, but in a more weakly manner, by trends of the atmospheric circulation. Other factors may then gain in relative relevance, explaining a larger portion of the variations in decadal trends.

In this study, we analyse long-term sea-level and climate records with the aim of explaining the observed variability of the decadal and multidecadal sea-level trends in the Baltic Sea. We further investigate whether or not the same mechanisms that

have been invoked to explain the interannual variations of Baltic sea-level are also responsible for the variability of the decadal sea-level trends.

For this purpose, and in contrast to most previous studies that focused on the interannual variations of sea-level, we statistically analyse sea-level and climate decadal trends. The analysis is carried out for each season separately. In this

analysis, we characterize the spatial coherency of the variations of decadal trends across the Baltic Sea and try to identify the connection between the variations of decadal sea-level trends and the variations of climate trends.

A series of studies have shown that an important part of the interannual to decadal variations of sea-level in the semi enclosed Baltic Sea result from atmospheric forcing, mostly from the wind (e.g. Heyen et al., 1996; Andersson, 2002;

Kauker and Meier, 2003; Chen and Omstedt, 2005; Hünicke and Zorita, 2006). The Baltic Sea is located in a region of predominantly westerly winds and it is connected to the North Sea by narrow straits. Both factors cause Baltic Sea level to be very sensitive to the regional atmospheric circulation. An important pattern of atmospheric circulation in this region at seasonal time scales is the North Atlantic Oscillation (NAO). Thus, many of the studies mentioned above have explored the statistical link between the NAO and the Baltic Sea level.

Although the NAO is an important factor modulating long-term (interannual to decadal) sea-level in the semi-enclosed Baltic Sea, its influence is not so strong in seasons other than winter. Additionally, the link between the NAO and the Baltic Sea level is spatially very heterogeneous even in wintertime, and has also displayed substantial decadal variations in the last two centuries (e.g. Andersson, 2002; Jevrejeva et al., 2005; Hünicke et al., 2015).

By using sea level pressure (SLP), sea surface temperature and precipitation observational time series for the winter and

summer seasons, Hünicke and Zorita (2006) concluded that precipitation and air temperature together with the SLP - including NAO SLP pattern- significantly modulate the sea-level variability on decadal time scales. They also showed that the influence of precipitation and temperature has a stronger effect on sea-level variations in summer than in winter. They suggested that sea-level variations are influenced by different factors in winter and summer seasons. Hünicke et al. (2008) identified that sea-level variations at central and eastern Baltic Sea are well described by the SLP alone, but, that area-

averaged precipitation may modulate the decadal sea-level variations in the southern Baltic.

Beyond the atmospheric forcing on the Baltic Sea level, we also explore here other possible mechanism that may be responsible for the decadal variability of Baltic Sea sea-level trends. Since the signal of the atmospheric forcing on Baltic sea-level can be very strong for some locations, we also apply a somewhat novel strategy to better identify the possible influence of the slowly-varying North Atlantic and North Sea sea-level. For this purpose, we set up a statistical model that

should capture the simultaneous link between atmospheric circulation and sea-level and then focus on the residuals of this statistical model, i.e. the part of variability of the sea-level that cannot be statistically explained by the simultaneous atmospheric circulation.

These atmospheric predictors in this statistical model are based on a Principal Component Analysis (PCA) of the SLP time series, retaining only the leading components that explain most of its variability. The residuals of this multivariate regression

analysis provide decadal sea-level trends that are not directly linked to the atmospheric forcing. We analyse these residuals and their connections to other atmospheric factors like precipitation in previous seasons and other oceanic factors like sea-level in the North Atlantic Ocean and the mode of North Atlantic sea-surface temperature known as the Atlantic Multidecadal Oscillation (AMO).

Besides, the Glacial Isostatic Adjustment (GIA) – which is a consequence of the Fennoscandian ice-sheet melting since the last Glacial Maximum– leads to long-term, secular negative trends of relative sea-level (referred to land) along the northern Baltic coast. The largest land uplift rates occur over the northern part of the Baltic Sea, and reach approximately 10 mm yr$^{-1}$. At the south coast of Baltic Sea the secular trend of vertical land movement is negative and around of -1 mm yr$^{-1}$ (Ekman, 1996; Peltier, 2004; Lidberg et al., 2010; Richter et al., 2011). Because our interest lies on the variability of climate-induced sea-level trends in the Baltic Sea region, we have to remove the GIA effect from sea-level time series.

The gliding trends of all sea-level and climatic observational records; tide gauge and sea surface height anomalies (SSHA) for sea-level, and the AMO-index, the NAO-index, the SLP fields, near-surface air temperature, precipitation are computed over running 11-year windows. We had one important advantage by carrying out the analysis on decadal gliding trends. This advantage is that the trends deduced from absolute (SSHA) and relative (tide gauge) can be more easily compared, since the GIA does not affect the calculation of the 11-year gliding trends since the GIA-induced trend of the Baltic Sea level does not presumably vary on time scales of one century,

In this study, new relative sea-level tide gauge data sets provided by the Technical University of Dresden are used. These data sets are a part of the Travemünde, Wismar, Warnemünde, Sassnitz, Swinoujscie, Kolobrzeg records and are the complete time series for the Marienleuchte, Barth, and Greifswald records.

Our research objectives can be summarised in the following questions: (1) How do long-term trend relationships between sea-level and climatic factors vary seasonally and spatially? (2) Apart from the effect of atmospheric circulation, is there any other underlying factor that modulates the decadal trends of Baltic Sea level? (3) Is there any signature of the Atlantic Multidecadal Oscillation (AMO) on decadal sea-level trends in the Baltic Sea?

This study is organised as follows: The datasets and methodology are described in section 2 and 3, respectively. In section 4, we provide main outcomes of this study and compared the results. Section 5 presents several conclusions.

## 2 Data

We used the seasonal means of the following sea-level and climatic data sets.

### 2.1 Sea level data

We used two different sorts of sea level data set. We obtained relative sea-level data from 29 tide gauges considering the availability and geographical distribution of stations along the Baltic Sea coast. The tide gauge data were provided from different sources (Bogdanov et al., 2000; Ekman, 2003; Holgate et al., 2013; Permanent Service for Mean Sea Level (PSMSL), 2016; Technical University of Dresden (TUD)). The tide gauges with data sources are illustrated in Figure 1.

**Figure 1: Tide gauges with their sources: 1-Aarhus, Barth, Frederikshavn, Furuogrund, 5-Greifswald, Hamina, Helsinki, Hirtshals, Kemi, 10-Klaipeda, Kolobrzeg, Kronstadt, Kungsholmsfort, Landsort, 15-Marienleuchte, Oslo, Pietarsaari, Ratan, Rauma, 20-Sassnitz, Slipshavn, Smogen, Stockholm, Swinoujscie, 25-Travemünde, Tregde, Visby, Warnemünde, Wismar (stations**

The tide gauge time series contain data gaps. Here, we considered the seasonal means and computed the 11-year gliding trends for each season only when 80% of time series (9 time steps) were available. The time coverage of the used tide gauge records is displayed in Figure 2.

**Figure 2: Time coverage of the tide gauge observations (monthly means). Numbers on the y-axis refer to the station number defined in Figure 1. Time series are coloured for better visibility.**

Together with the tide gauge observations, we used SSHA time series which are reconstructions of sea-level based on statistical processing of tide gauges and satellite altimetry observations to achieve a longer temporal coverage than provided by the satellite data alone. This reconstruction spans the period covering from 1950 to 2008. To reconstruct sea-level fields, satellite altimetry derived cyclostationary empirical orthogonal functions are combined with tide gauge observations. This reconstructed sea-level data was obtained from Jet Propulsion Laboratory (JPL) Physical Oceanography Distributed Active Archive Center (PO.DAAC) and developed by the University of Colorado (Hamlington et al., 2011).

## 2.2 Climatic data sets

Climatic Data Sets include the North Atlantic Oscillation (NAO) index, Atlantic Multidecadal Oscillation (AMO) index, sea level pressure (SLP), near surface air temperature, precipitation and evaporation. The NAO-index is derived from the difference between the normalized sea-level-pressure (SLP) in Gibraltar and Reykjavik. The index covers the period from 1821 to 2012 (Jones et al., 1997).

The AMO-index is computed based on the area weighted average of sea surface temperature over the North Atlantic between $0°$ to $70°$ N (Enfield et al., 2001). The AMO-index used here was provided by National Oceanic and Atmospheric Administration/Physical Sciences Division (NOAA/PSD), covering the period 1856-2015.

The SLP data are $5°x5°$ gridded Northern Hemisphere monthly means from 1899 to present and are provided by the National Centre for Atmospheric Research (NCAR; Trenberth and Paolino, 1980). We used the domain between $70°$W -$40°$E and $30°$N - $90°$N in this study.

We used the combined HadCRUT4 land and marine surface temperature anomalies from CRUTEM4 and HadSST3 with $5°x5°$ gridded monthly means, starting in 1850 and continuing until present and covering the area of $60°$W – $40°$E and $32°$N – $70°$N (Morice et al., 2012).

We used two different precipitation data sets. One was the gridded $0.5°X0.5°$ monthly means from 1900 to 2012 in the geographical window $20°$W-$40°$E and $48°$N-$70°$N, obtained from the Climatic Research Unit (CRU; Harris et al., 2014; Trenberth et al., 2014). This data set represents precipitation only over land.

The second precipitation data set was monthly means from the meteorological reanalysis of the National Center for Environmental Protection/National Center for Atmospheric Research (NCEP/NCAR; Kalnay et al., 1996; Kistler et al.,

2001). The reanalysis data set has a spatial resolution of 192x94 points with T62 Gaussian grid covering the area (88.542° N – 88.542° S) latitudes and (0° E – 358.125° E) longitudes over the Earth's surface. Here, we considered the period from 1948 to 2012 for the area covering the drainage basin of Baltic Sea. Also, surface evaporation data set is derived from the surface latent heat flux from the NCEP/NCAR reanalysis data. Precipitation and evaporation of the meteorological reanalysis also include ocean areas.

## 3 Methodology

The tide gauge records contain the secular signal due to global climatic change and the postglacial land uplift, which cause a long-term trend in the sea-level observations. As it is mentioned in the introduction, our focus is the analysis of the variability of decadal sea-level trends over the last century. Except for the reanalysis and SSHA data sets, the analysis is carried out for the period 1900-2012.

After selecting the sea-level and climatic data sets with their time ranges, each season is treated separately. We computed the 11-year gliding trends for every season (Winter-DJF, Spring-MAM, Summer-JJA and Autumn-SON), requiring at least 80% availability of data for each single gliding trend computation. The gliding trend for each 11-year window is estimated through linear regression on time, as described by Equation (1).

$$SL_i = at_i + b, \tag{1}$$

The term $SL_i$ denotes the observed value of sea-level, $t_i$ is the ith year within the 11-year window, $a$ is the trend of sea-level with respect to time and b is the y-axis intercept. We used ordinary-least-squares estimation for the linear regression analyses.

In the following step, we first applied PCA to the SLP gliding trend time series to capture the leading five principal vectors of the SLP fields representing most of the variance of the SLP gliding trends in the geographical area between 70°W -40°E and 30°N - 90°N. Afterwards, we conducted a multivariate linear regression analysis with the leading five principal components of the SLP trends as predictors and the tide gauge gliding trends as predictands.

A multiple linear regression model is defined in Equation (2).

$$Y(t) = ß_0 + \sum_{i=1}^{5} ß_i \, PCSLP_i(t) + e_j(t), \tag{2}$$

Herein, $PCSLP_i(t)$ are the time series of the $i^{th}$ principal component of SLP-trend and $ß_i$ are the regression coefficients of the leading principal component. Y is the time series of gliding trend anomaly for each tide gauge. The error term, $e_j(t)$, is the vector of sea-level trend residuals which cannot be linearly described by the first five principal component vectors of the SLP trends. We used those residuals assuming that they contain the variability of sea-level trends not caused by the simultaneous atmospheric (SLP) forcing.

We also tested whether inverting the ordering of computation of the decadal SLP trends and PCA filtering could influence the results of the regression model and estimation of the residuals. The differences obtained were very small. A correlation is

considered significant when it passes the 95% level under the usual assumptions of normally distributed and temporally uncorrelated variables. For this estimation we used the usual t-distribution test.

## 4 Results

The NAO is widely known as a major atmospheric factor modulating the sea-level in the Baltic Sea region on interannual time scales. To confirm that this link is also valid for the decadal trends, we show in Figure 3 the correlation pattern between the decadal trend of the NAO-index and the decadal trends of 29 tide gauges in the Baltic Sea in wintertime.

**Figure 3: Correlations of decadal gliding trends between the NAO-index and 29 tide gauges for wintertime (1900-2012). The 95% significance level is ±0.19 for this record length.**

The correlation pattern in Figure 3 shows that the link between the trends of the NAO and trends of tide gauges is very heterogeneous in space. Particularly, this correlation pattern indicates a strong variation from north to south of the Baltic Sea basin. This result is consistent with the findings of former studies on the interannual correlation between the NAO and tide gauges (e.g. Hünicke and Zorita, 2006). The stations along the southern part of the Baltic Sea have a weak link to the NAO, and the effect of the NAO becomes stronger towards the north of the Baltic Sea.

We now investigate whether this spatially heterogeneous link between the NAO and the tide gauges in wintertime is also reflected in small inter-correlations between the sea-level trends derived in the different areas of the Baltic Sea. The rationale behind this investigation is that if the NAO is the major factor modulating the sea-level trends and the effect of the NAO on sea-level is spatially heterogeneous, then the sea-level trends should also be only weakly related. For this purpose, we select one tide gauge from the central Baltic (Stockholm), and the other one (Warnemünde) as representative station of the Baltic proper and of the south coast of the Baltic Sea for further analysis. We then calculate the correlations between each of these two tide gauges and the trends derived from reconstructions (SSHA) including the satellite altimetry observations over the whole Baltic Sea. The correlation patterns are illustrated in Figure 4.

**Figure 4: Correlations of decadal gliding trends between selected (Stockholm-left and Warnemünde-right) tide gauges and SSHA grids for all seasons shown in the order of winter-first row, spring-second row, summer-third row, autumn-last row (period: 1950-2008). The 95% significance level is ±0.25 for this record length.**

Although the NAO has non-uniform link to the Baltic Sea region, the correlations between SSHA and each of the two representative stations seem to be very similar. This indicates that even though the tide gauges along the southern coast of the Baltic Sea have weak connections to the NAO, sea-level trends in the Stockholm and Warnemünde tide gauges are strongly correlated to each other and with decadal sea-level trends of SSHA reconstructions in the Baltic Sea. One explanation that we pursue further in this analysis is that another factor, independent of direct atmospheric forcing encapsulated by the NAO, is more strongly responsible for the spatial homogeneity of the sea-level decadal trends.

To identify this factor, we statistically filter the influence of the atmospheric forcing on the decadal sea-level trends. This is accomplished by a regression model that uses as predictors the leading five principal components of the SLP trends that

explain 89%, 81%, 78% and 79% variance of the SLP trends for winter, spring, summer and autumn, respectively, as indicated in Equation (2). It should be noted that these are explained variances of the SLP trends, and not the sea-level variance that can be explained by SLP trends. After that principal component analysis of the SLP field trends, we implemented a multivariate regression where those principal components of the SLP trends were used as predictor and decadal trends of sea-level records were predictand. The residuals of the multivariate regression analysis for the tide gauges were used as new decadal trends which are presumably free of the direct atmospheric forcing.

We then compute the correlations between the residuals of the two tide gauges (Stockholm and Warnemünde) and residuals of the rest of the nine tide gauges (Ratan, Stockholm, Helsinki, Smogen, Kungsholmsfort, Sassnitz, Travemünde, Wismar, Warnemünde) in the Baltic Sea. We consider these tide gauges, nine tide gauges in total, to be representative of the Baltic Sea, reasonable span the Baltic Sea region and have long records. Figure 5 represents the correlation patterns between the two selected tide gauges and the other tide gauges, for both the decadal trends and the residuals resulting from filtering the effect of the SLP trends.

**Figure 5: Correlation between sea-level at two representative tide gauges (Stockholm and Warnemünde) and at the rest of the tide gauges based on decadal trends. First (last) two columns present the correlation between Stockholm (Warnemünde) and the other tide gauges. For each tide gauge, the left (right) column shows the correlations obtained with the observed (residual) record. The maps are ordered in the line of winter-first row, spring-second row, summer-third row, autumn-last row. The 95% significance level is ±0.19 at the 95% for this record length. Note that correlation scale ranges from 0.2 to 1.**

Figure 5 illustrates that the correlations between stations tend to become stronger after removing the atmospheric effect from the decadal sea-level trends. For example, before removing the atmospheric effect in wintertime, the correlation between Warnemünde and Stockholm decadal trends was 0.72. However, it increased to 0.89 after removing the atmospheric signal from both stations.

To confirm that the atmospheric signal in decadal sea-level trend anomalies is effectively removed and to show, at the same time, that the similarity of the trend anomalies increases after this removal, we examined the relationships of two representative tide gauges with both the SLP field and the near-surface air temperature in wintertime. The reason for including here the near-surface temperature in the analysis is that in wintertime temperature variations are very strongly affected by advection by the atmospheric circulation. A lack of (indirect) correlation between sea-level and air-temperature would also indicate that the atmospheric signal has been removed from the sea-level trends. The results of this analysis are represented in the Figures 6 and 7, respectively.

**Figure 6: The correlation patterns between SLP fields and Stockholm and Warnemünde before (top) and after (bottom) removing the atmospheric signal in wintertime. The areas indicating significant correlations are delineated with contour lines. The 95% significance level is ±0.19 for this record length.**

In Figure 6, the two maps in the top row show the correlation patterns between decadal trends of the tide gauges and of the SLP field. We see that the stations Stockholm and Warnemünde are heterogeneously connected to the SLP field. However, the patterns regarding the correlations of decadal trend anomalies between SLP fields and residuals of sea-level

trend anomalies (the two maps in the bottom row) indicate very similar variations of the Stockholm and Warnemünde tide gauges. In the next step, we replaced the SLP fields by the near-surface air temperature anomalies in order to display the correlation patterns between the two tide gauges and the air temperature. Here, we explore the indirect correlation between SL and temperature, which is triggered by the atmospheric circulation, in wintertime. It should be noted that the NAO is correlated with both SL and air temperature in this region. This analysis is not referring a causal relationship between winter temperature and SL in the Baltic Sea. It is rather carried out in order to show that atmosphere driven temperature is removed from the decadal sea-level trends by using multivariate regression analysis.

The results of that connection between air temperature decadal trends and sea-level decadal trends are illustrated in Figure 7.

**Figure 7: The correlation patterns between air temperature and the sea-level records at Stockholm and Warnemünde before (top) and after (bottom) removing the atmospheric signal from the sea-level records in wintertime. The areas indicating significant correlations are delineated with contour lines. The 95% significance level is ±0.19 for this record length.**

The Figures 6 and 7 confirm that the atmospheric signal is removed from the decadal tide gauge trend anomalies. Moreover, they show that the correlation patterns of Stockholm and Warnemünde residuals are very similar in terms of their correlations to near-surface air temperature and to the SLP time series.

In summary, the results suggest that SLP, and therefore mean seasonal wind, forcing has a spatially heterogeneous effect on different locations and this occurs for all four seasons. After removing the effect of the SLP from the tide gauge 11-year gliding trend time series, most of these correlations become more clear and stronger. This suggests that there is an underlying factor modulating sea-level trends uniformly through the whole Baltic Sea basin.

Here, it should be noted that those trend residuals account for a considerable amount of the variability of sea-level trends. For example, the sea-level trend residuals explain 41% of the total trend variance of Stockholm in wintertime. This also indicates that 59% variance of sea-level trends are explained by the first five principal components of SLP patterns in this season. In addition, those residuals of sea-level trends show substantial deviations between 9.5mmyr$^{-1}$ and -21.1 mmyr$^{-1}$ over the period 1900-2012. Associated sea-level trends of the Stockholm station range from 23.7 mmyr$^{-1}$ to -33.1mmyr$^{-1}$ over the same period. To explore the nature of this underlying effect causing a more uniform variability in the sea-level trend residuals, we investigated two possible physical mechanisms. One is the role of precipitation in the Baltic Sea catchment area. The other factor is the role of the low-frequency variability in the North Atlantic, as described by the Atlantic Multidecadal Oscillation (AMO)(Enfield et al., 2001).

The results concerning precipitation are represented in Figure 8. Since precipitation over the catchment area of the Baltic Sea would affect sea-level only after some lag, we investigated, for each season separately, the correlations between precipitation in the previous season and tide gauge residuals. This correlation is based on decadal trend time series.

The results, for the CRU precipitation data set, are represented in Figure 8.

**Figure 8: The correlation patterns between the decadal sea-level trend residuals and the area averaged CRU precipitation decadal trends in the previous season over the Baltic Sea catchment area for the period 1900-2012. The left (right) panels show the results of Stockholm (Warnemünde) station. The areas indicating significant correlations are delineated with contour lines.**

Since precipitation is strongly controlled by the atmospheric circulation, we also investigated the link between the sea-level trends and SLP trends in the previous season. The patterns are shown in Figure 9.

**Figure 9: The correlation patterns between decadal gliding trends of the sea-level residuals and decadal SLP trends in the previous season. The left (right) panels show the results of Stockholm (Warnemünde) station. The areas indicating significant correlations are delineated with contour lines. The 95% significance level is ±0.19 for this record length.**

The figures display that, in addition to the atmospheric forcing, there is a lagged contribution of precipitation to the decadal sea-level trends in the Baltic Sea. This contribution seems to be strong on the decadal sea-level trend variability, except for the spring season.

To further quantify the effect of the precipitation on the following season, we also used reanalysis precipitation data in addition to the CRU precipitation data. In contrast to the CRU precipitation data set covering only land, the reanalysis data covers both ocean and land, but with coarser spatial resolution. Besides, the temporal coverage of reanalysis precipitation data is shorter, starting in 1948. Considering the drainage basin of the Baltic Sea, the spatial means of two data sets are computed in order to examine the covariability between precipitation and sea-level residuals. Table 1 shows the results of correlation analysis between precipitation (both from CRU and NCEP/NCAR) together with the freshwater flux (P-E) (from NCEP/NCAR data) and residuals of sea-level in terms of the decadal gliding trend variations. In Table 1, the climatic factors precipitation and freshwater flux are correlated to the lagged seasonal mean sea-level trend residuals.

**Table 1: The correlations of decadal trends between sea-level at the Stockholm and Warnemünde stations and area-averaged precipitation (PRE) from CRU or NCEP/NCAR in the previous seasons. Additionally, the correlations between decadal trends of freshwater flux (Precipitation-Evaporation) field means and lagged sea-level trend residuals are shown. The 95% significance levels are r>0.26 for CRU data set and r>0.19 for NCEP/NCAR data set. The significant correlation coefficients are marked with (\*) symbols.**

Table 1 indicates that precipitation has a considerable lagged effect on decadal trends of sea-level variations, once the direct SLP forcing has been subtracted from the sea-level records. For instance, taking the results of both precipitation data sets into account, summer season precipitation implies relatively strong linkage to the residuals of sea-level decadal trend variations, reaching to r=0.56 in autumn.

Concerning the results only from precipitation of the reanalysis data set, it is shown that there are significant connections between winter precipitation and summer sea-level residuals, between summer precipitation and autumn sea-level residuals and between autumn precipitation and winter sea-level residuals for the selected two stations. However, the results of reanalysis precipitation time series do not imply a significant connection between spring precipitation and sea-level residuals in any season. Considering the freshwater flux effect in the analysis, it should be mentioned that evaporation in winter and autumn contributes to the precipitation explained variance of sea-level trends in the summer and winter seasons, respectively, for both stations. The results derived using the CRU precipitation indicate that, on the one hand, winter precipitation affects decadal trends of sea-level in the spring, that summer precipitation contributes to sea-level decadal trends in autumn for both stations. Besides, autumn season precipitation seems to explain a part of variation of Warnemünde

sea-level residuals in winter. On the other hand, precipitation decadal trends of spring season do not have a significant connection to the decadal trends of sea-level residuals.

To examine other possible large-scale factors on sea-level trends, we investigated the potential influence of the North Atlantic sea-surface temperature anomalies in the form of the AMO-index. The 11-year standardized gliding trend anomalies of sea-level residuals in wintertime and the standardized trends of the AMO-index for each season are represented in Figure 10.

**Figure 10: The standardized (unitless) decadal gliding trend time series of the AMO-index (black) and tide gauge residuals (blue-Stockholm, red-Warnemünde)**

This particular analysis suggests very weak relations between the AMO-index and residuals of sea-level trend anomalies. The strongest correlation in all seasons was 0.2. These results indicated that there is no significant contribution of the AMO related factor to decadal sea-level trend residuals in the Baltic Sea region.

## 5 Conclusions

We statistically investigated the variability of the decadal trends in the Baltic Sea over the period 1900-2012 and explored various physical factors that may explain this variability. The decadal trends of the Baltic Sea level are influenced by the SLP and therefore by the wind forcing, as in the case of interannual variations. Those SLP fields explain considerable (i.e. 59% in wintertime) amount of decadal sea-level trends. However, this influence is spatially heterogeneous, with a stronger effect in the northern and a weaker one in the southern Baltic Sea. This contrasts with a rather spatially homogeneous variation of the decadal sea-level trends in the Baltic Sea, which implies that SLP alone is likely not the sole factor that drives the variations of the decadal sea-level trends.

To identify this underlying factor, we explored the role of precipitation and of the Atlantic Multidecadal Oscillation (AMO). Precipitation in the previous season over the Baltic Sea catchment area seems to be a robust candidate to explain the variations of the decadal sea-level trends in the summer and autumn seasons, as well as partly in wintertime, but its role is much weaker in the spring season. Evaporation in the winter and autumn seasons contributes to that lagged connection between precipitation and sea-level trend residuals. The lagged effect of precipitation is rather homogeneous for all Baltic Sea tide gauges.

We could also not identify a clear contribution of the North Atlantic Ocean to variations of the sea-level trends in the Baltic Sea in any of the four seasons. This is likely due to the strong regional forcing of the sea-level trends at decadal time scales.

Some implications can be derived from these conclusions. Although the future anthropogenic sea-level trends in the future in the North Atlantic and probably area-averaged sea-level will roughly follow the global sea-level rise caused by anthropogenic warming, the Baltic Sea, the decadal variations around this trend, linked to natural climate variations, may be different in the North Atlantic and the Baltic Sea and within the Baltic Sea itself. The effects of the trends in the SLP are spatially heterogeneous, and most important in wintertime. The effect of precipitation trends, assuming that the same

mechanisms remain unaltered in the future, may affect some seasons more strongly than others, and therefore the seasonal decadal trends in sea-level may also more strongly diverge in the future if the large-scale hydrological cycle is intensified in a future climate. This effect may be smoothed if the impact of river regulation becomes stronger in the future. Therefore, detection and attribution studies focusing on identifying the anthropogenic signal (due to greenhouse gas forcing) on the

Baltic sea-level would need to consider the spatial and seasonal heterogeneity of the atmospheric and oceanic drivers on decadal sea-level variations

The role of the Atlantic Multidecadal Oscillation appears also to be limited in the Baltic sea level variability in the historical record, compared to other regional drivers. This implies that in the future decades the variations around the secular sea-level rise in both seas would continue to appear uncoupled. This could have some implications, for instance, for the rate of water

exchange between the North Sea and the Baltic Sea, and therefore explain part of the decadal variability in the frequency of North Sea water inflows into the Baltic Sea. This impact has not been further explored here.

## Acknowledgements

This study was supported by the Deutsche Forschungsgemeinschaft (DFG) through the CliSAP project. The tide gauge data was obtained from the PSMSL, and some parts of tide gauge data sets were from Martin Ekman, and Andreas Groh from

TUD and FGD. We also thank NASA/JPL PO.DAAC and University of Colorado for the reconstructed SSHA time series. We are also grateful to the institutes NOAA, CRU, NCEP and NCAR for the climatic data sets used in this study.

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

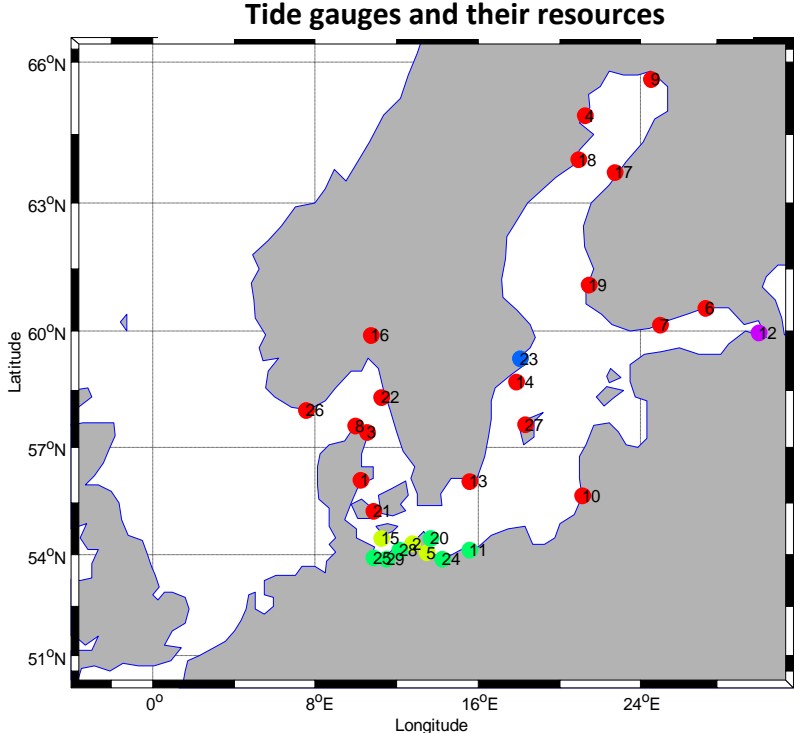

**Figure 1: Tide gauges with their sources: 1-Aarhus, Barth, Frederikshavn, Furuogrund, 5-Greifswald, Hamina, Helsinki, Hirtshals, Kemi, 10-Klaipeda, Kolobrzeg, Kronstadt, Kungsholmsfort, Landsort, 15-Marienleuchte, Oslo, Pietarsaari, Ratan, Rauma, 20-Sassnitz, Slipshavn, Smogen, Stockholm, Swinoujscie, 25-Travemünde, Tregde, Visby, Warnemünde, Wismar (stations are ordered alphabetically). (Red: PSMSL; Purple: Bogdanov et al.; Yellow: TUD; Green: PSMSL and TUD; Blue: Ekman and PSMSL)**

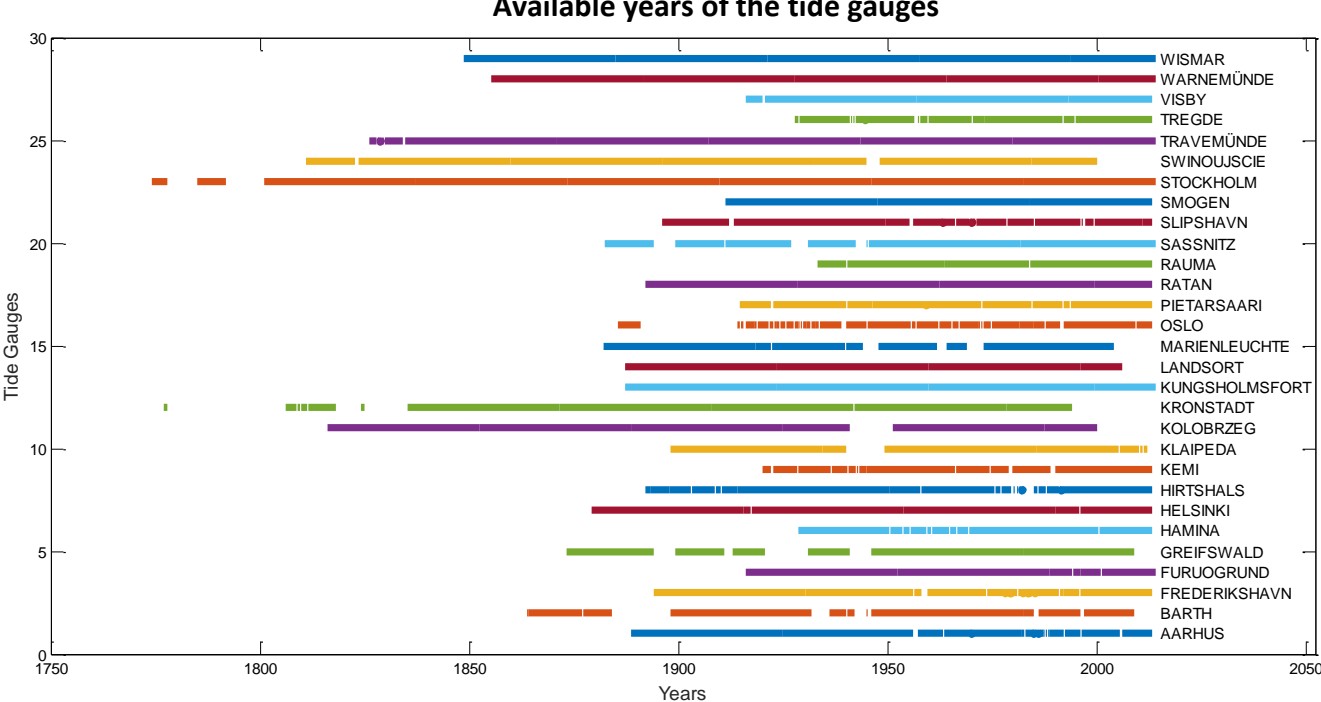

**Figure 2: Time coverage of the tide gauge observations (monthly means). Numbers on the y-axis refer to the station number defined in Figure 1. Time series are coloured for better visibility.**

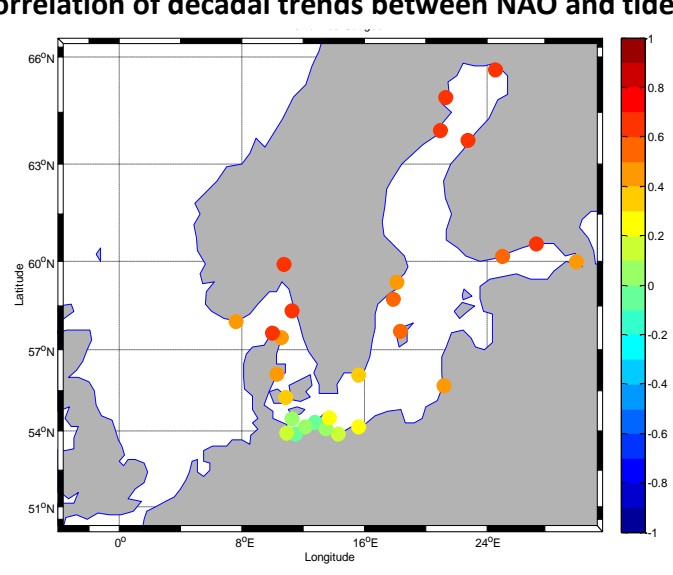

**Figure 3: Correlations of decadal gliding trends between the NAO-index and 29 tide gauges for wintertime (1900-2012). The 95% significance level is ±0.19 for this record length.**

**Correlation patterns of decadal trends between sea surface height anomalies and tide gauges**

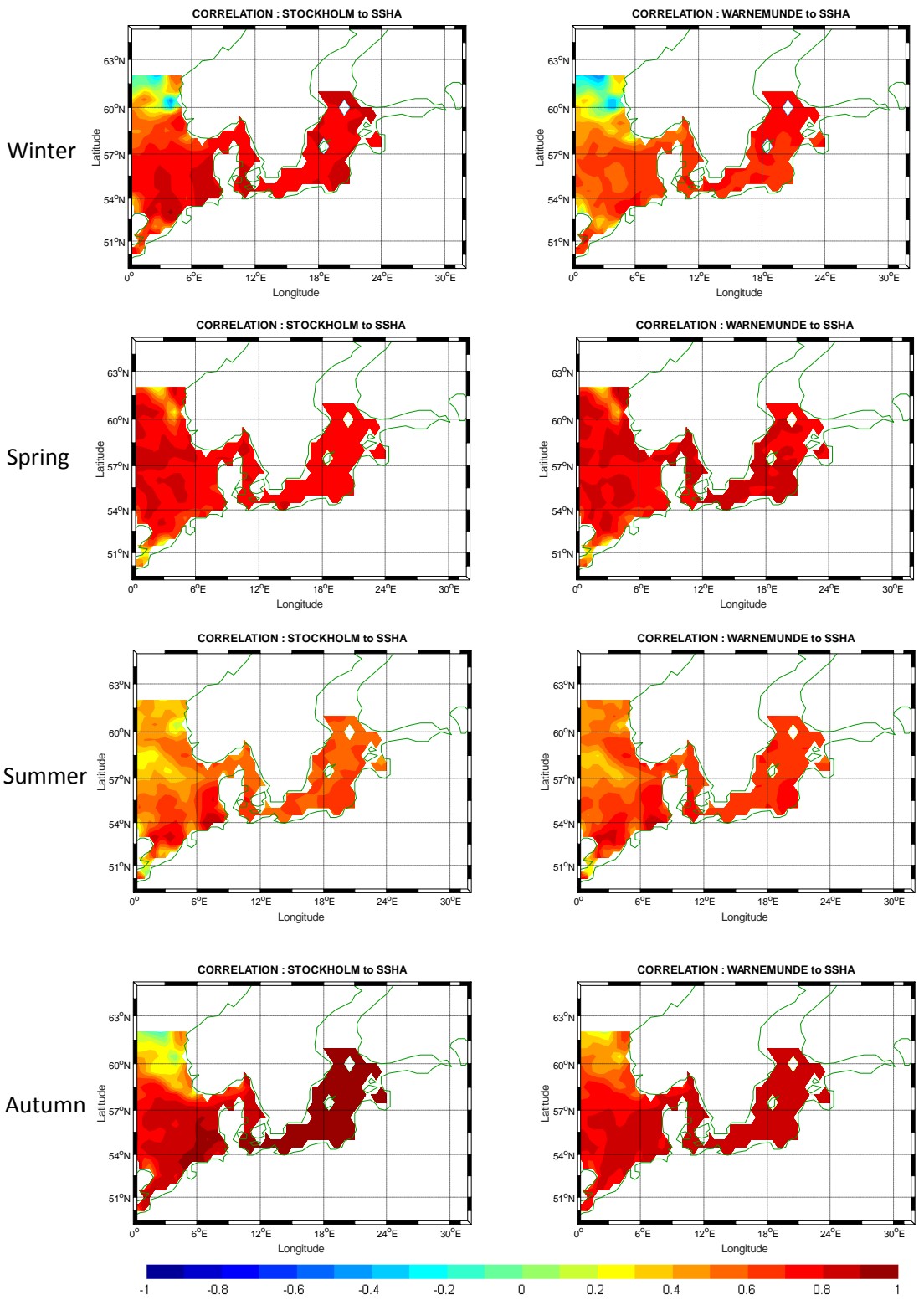

**Figure 4: Correlations of decadal gliding trends between selected (Stockholm-left and Warnemünde-right) tide gauges and SSHA grids for all seasons shown in the order of winter-first row, spring-second row, summer-third row, autumn-last row (period: 1950-2008). The 95% significance level is ±0.25 for this record length.**

# Tide gauge correlations based on decadal gliding trends

**Figure 5: Correlation between sea-level at two representative tide gauges (Stockholm and Warnemünde) and at the rest of the tide gauges based on decadal trends. First (last) two columns present the correlation between Stockholm (Warnemünde) and the other tide gauges. For each tide gauge, the left (right) column shows the correlations obtained with the observed (residual) record. The maps are ordered in the line of winter-first row, spring-second row, summer-third row, autumn-last row. The 95% significance level is ±0.19 at the 95% for this record length. Note that correlation scale ranges from 0.2 to 1.**

## Correlation patterns of decadal trends between SLP fields and tide gauges in wintertime

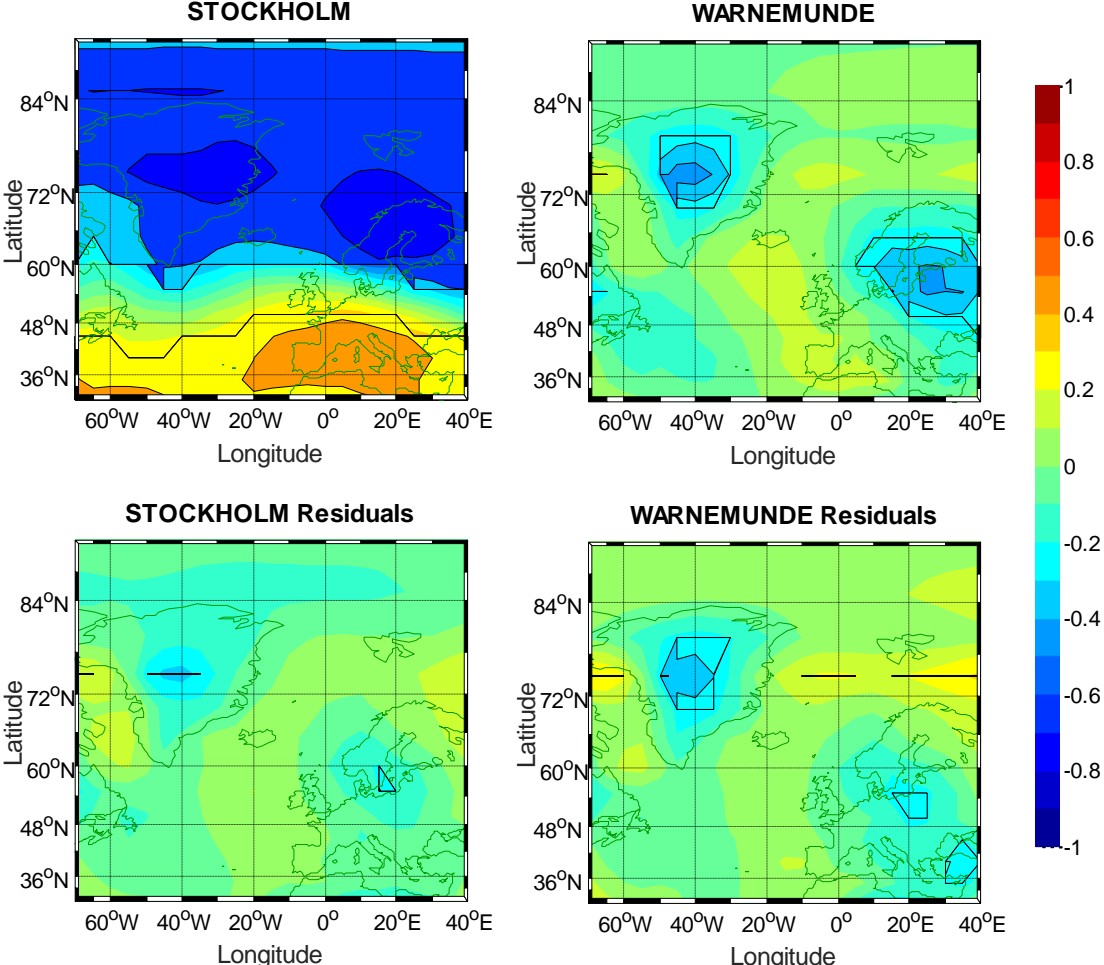

**Figure 6: The correlation patterns between SLP fields and Stockholm and Warnemünde before (top) and after (bottom) removing the atmospheric signal in wintertime. The areas indicating significant correlations are delineated with contour lines. The 95% significance level is ±0.19 for this record length.**

## Correlation patterns of decadal trends between air temperature and tide gauges in wintertime

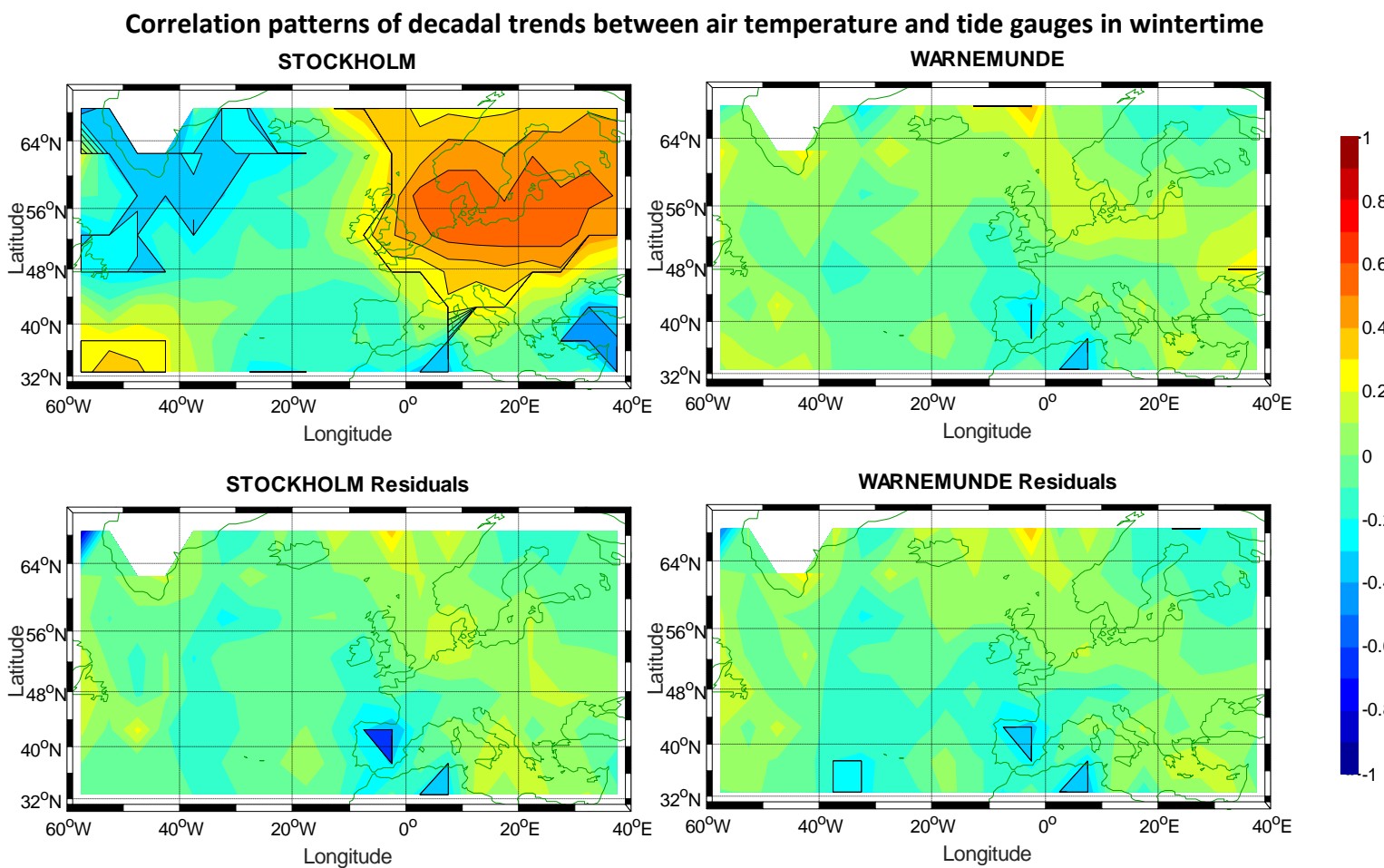

**Figure 7:** The correlation patterns between air temperature and the sea-level records at Stockholm and Warnemünde before (top) and after (bottom) removing the atmospheric signal from the sea-level records in wintertime. The areas indicating significant correlations are delineated with contour lines. The 95% significance level is ±0.19 for this record length.

# Correlation patterns of decadal trends between precipitation and lagged sea-level trend residuals

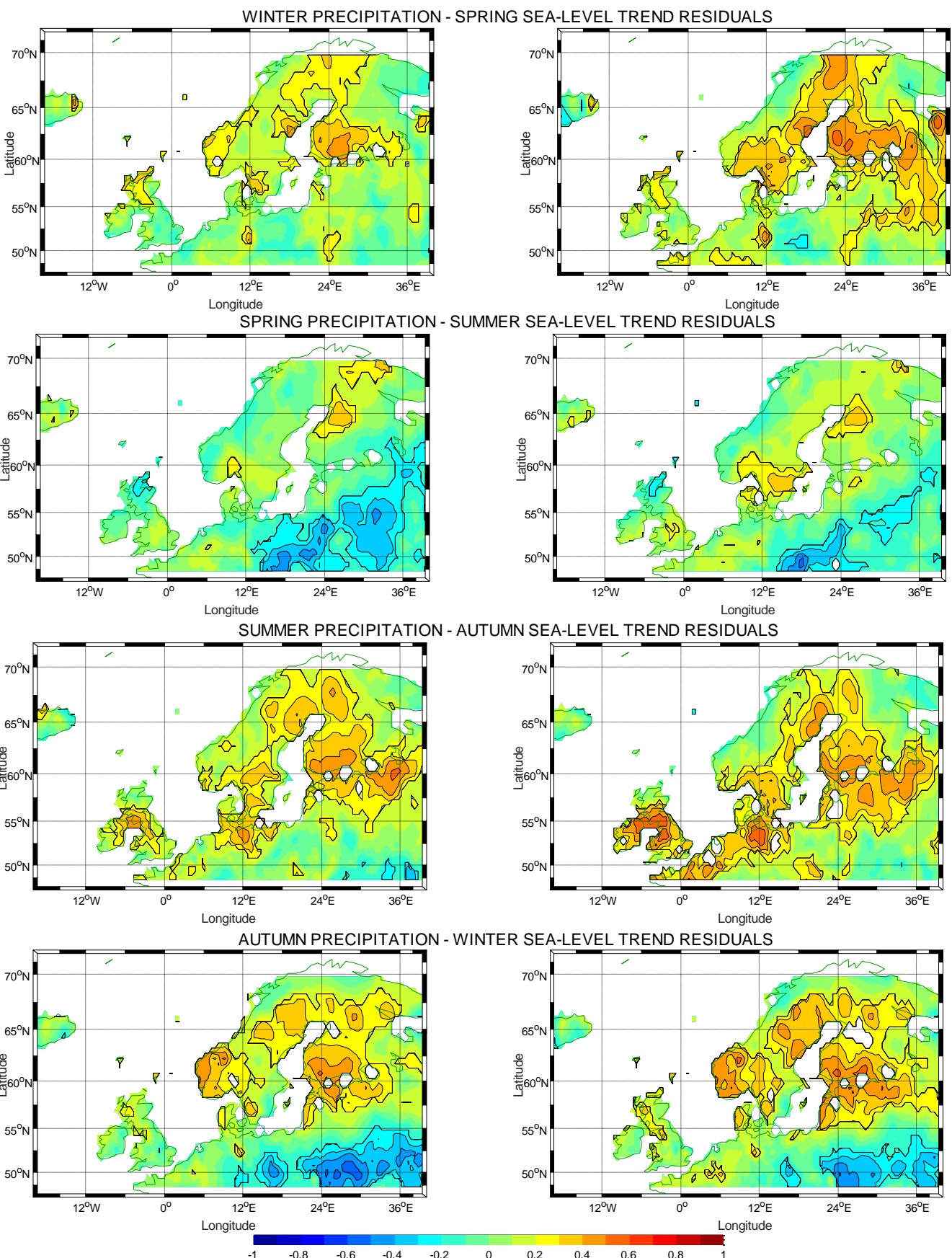

**Figure 8: The correlation patterns between the decadal sea-level trend residuals and the area averaged CRU precipitation decadal trends in the previous season over the Baltic Sea catchment area for the period 1900-2012. The left (right) panels show the results of Stockholm (Warnemünde) station. The areas indicating significant correlations are delineated with contour lines.**

# Correlation patterns of decadal trends between SLP fields and lagged sea-level trend residuals

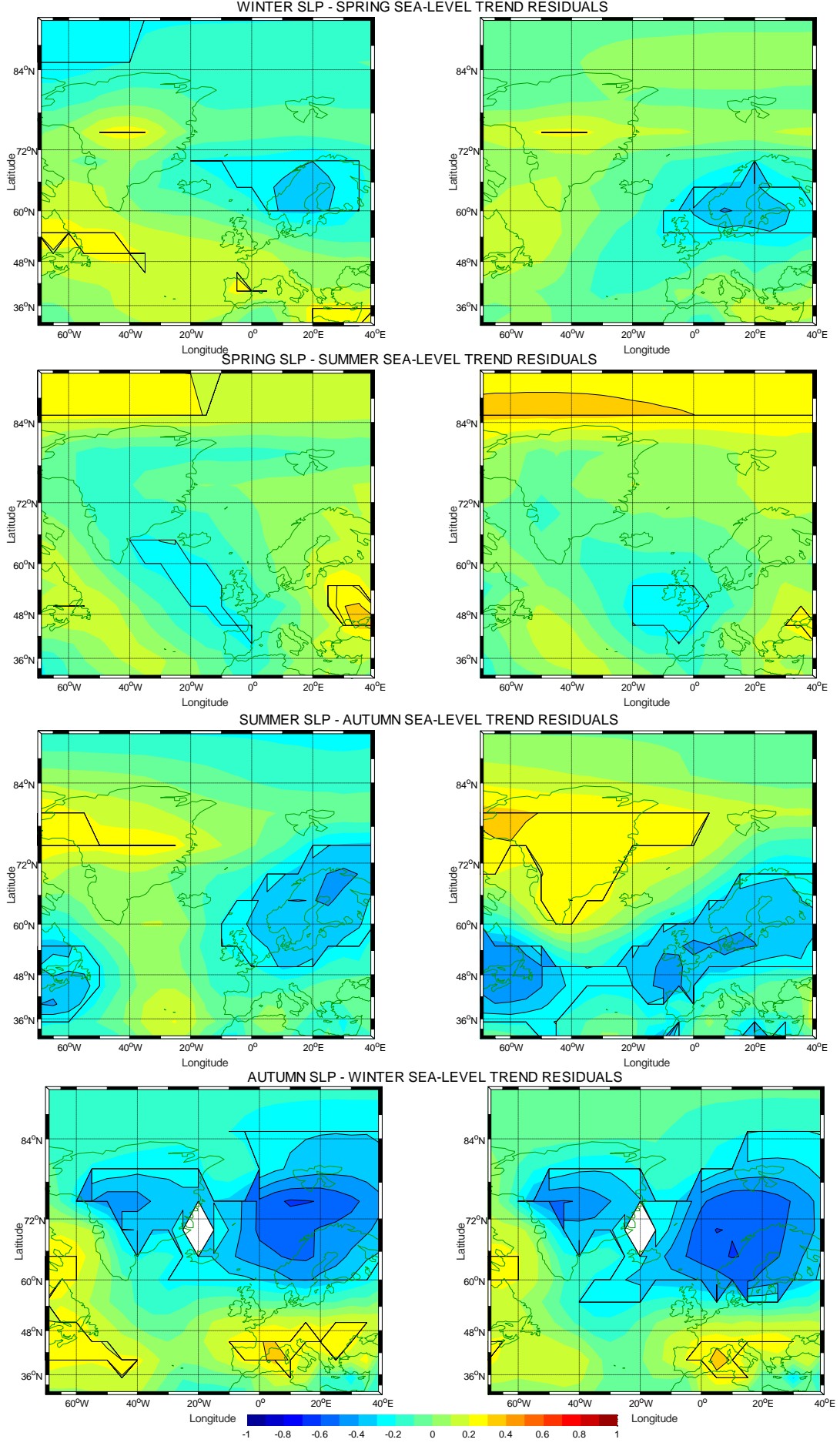

**Figure 9: The correlation patterns between decadal gliding trends of the sea-level residuals and decadal SLP trends in the previous season. The left (right) panels show the results of Stockholm (Warnemünde) station. The areas indicating significant correlations are delineated with contour lines. The 95% significance level is ±0.19 for this record length.**

**Table 1: The correlations of decadal trends between sea-level at the Stockholm and Warnemünde stations and area-averaged precipitation (PRE) from CRU or NCEP/NCAR in the previous seasons. Additionally, the correlations between decadal trends of freshwater flux (Precipitation-Evaporation) field means and lagged sea-level trend residuals are shown. The 95% significance levels are r>0.26 for CRU data set and r>0.19 for NCEP/NCAR data set. The significant correlation coefficients are marked with (*) symbols.**

| Season | Station | | | | | | | |
|---|---|---|---|---|---|---|---|---|
| | Stockholm | | | | Warnemünde | | | |
| Winter | Lag0 | Lag1 | Lag2 | Lag3 | Lag0 | Lag1 | Lag2 | Lag3 |
| **Reanalysis FWF** | 0.26* | 0.02 | 0.54* | 0.08 | 0.20 | 0.08 | 0.42* | 0.24 |
| **Reanalysis (PRE)** | 0.14 | 0.11 | 0.48* | 0.04 | 0.05 | 0.15 | 0.29* | 0.21 |
| **CRU (PRE)** | 0.26* | 0.36* | 0.38* | 0.02 | 0.21* | 0.51 * | 0.19 | -0.01 |
| Spring | Lag-1 | Lag0 | Lag1 | Lag2 | Lag-1 | Lag0 | Lag1 | Lag2 |
| **Reanalysis FWF** | 0.16 | 0.03 | 0.06 | 0.05 | 0.06 | -0.09 | 0.16 | -0.09 |
| **Reanalysis (PRE)** | -0.01 | 0.10 | 0.00 | -0.06 | -0.13 | -0.03 | 0.10 | -0.19 |
| **CRU (PRE)** | -0.07 | 0.21* | -0.01 | 0.06 | 0.00 | 0.15 | 0.18 | 0.08 |
| Summer | Lag-2 | Lag-1 | Lag0 | Lag1 | Lag-2 | Lag-1 | Lag0 | Lag1 |
| **Reanalysis FWF** | 0.23 | -0.03 | 0.40* | 0.46* | 0.33 | 0.09 | 0.18 | 0.52* |
| **Reanalysis (PRE)** | 0.24 | -0.02 | 0.42* | 0.49* | 0.34* | 0.11 | 0.20 | 0.56* |
| **CRU (PRE)** | 0.10 | 0.03 | 0.29* | 0.49* | 0.19 | -0.03 | 0.14 | 0.52* |
| Autumn | Lag-3 | Lag-2 | Lag-1 | Lag0 | Lag-3 | Lag-2 | Lag-1 | Lag0 |
| **Reanalysis FWF** | 0.45* | -0.47 | 0.02 | 0.32* | 0.45* | -0.46 | 0.02 | -0.01 |
| **Reanalysis (PRE)** | 0.32* | -0.41 | -0.10 | 0.17 | 0.32* | -0.41 | 0.00 | -0.12 |
| **CRU (PRE)** | 0.16 | -0.17 | -0.16 | 0.16 | 0.25* | -0.15 | 0.02 | 0.08 |

# Decadal gliding trend time series of AMO-index and sea-level trend residuals

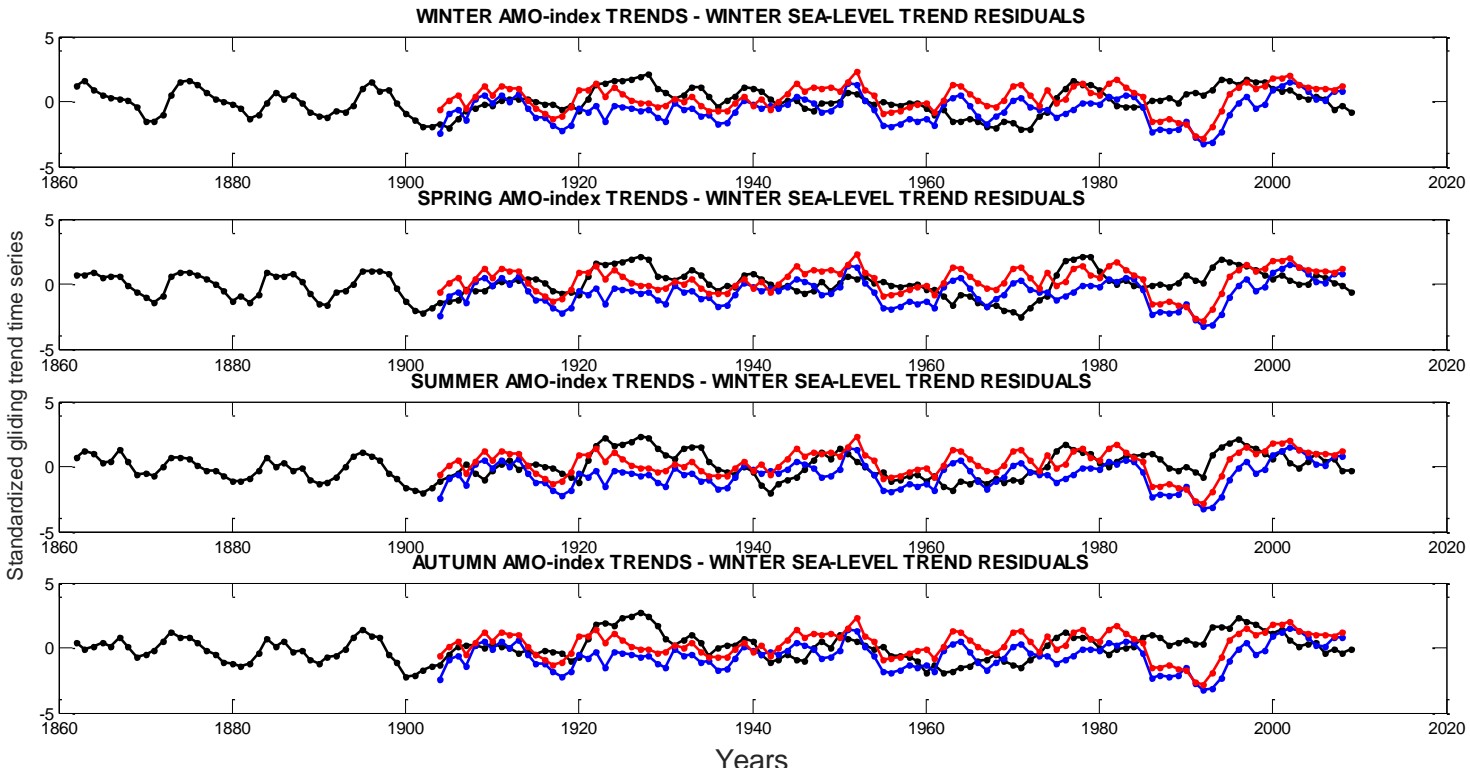

**Figure 10: The standardized (unitless) decadal gliding trend time series of the AMO-index (black) and tide gauge residuals (blue-Stockholm, red-Warnemünde).**