# Peer review of "Mechanisms of variability of decadal sea-level trends in the Baltic Sea over the 20th century"

_Earth System Dynamics, 2017_

## Referee Comment (RC1) · Anonymous Referee #1 · 13 Apr 2017

**Recommendation**

**Major revisions.**

**Synopsis**

The paper analyses the sea level (SL) variability in the Baltic Sea and its drivers. Sea level observations from 29 tide gauges, some of them going back to the eighteenth century, from around the Baltic Sea are used together with a satellite-based reconstruction of sea level for the whole Baltic that goes back to 1950. Observation-based data sets of SLP, precip, and temperature are used to investigate their relation to SL variations. The

paper focuses on longer time scales by considering the relation between decadal-scale trends of the variables, rather than the variables themselves.

SL variations on longer time scales are found to be highly connected to NAO (North Atlantic Oscillation) variations, with larger correlations in the northern than in the southern Baltic. Precipitation in the Baltic catchment also plays an important role for SL variability.

**Discussion**

The paper seems to be technically sound, but I miss an explanation of the relevance of the results. What are possible implications?

Furthermore, the presentation is often unclear. The list below gives some hints as to where the presentation of the work can be improved.

Together, I think that a major re-writing of the paper is necessary before it can be accepted.

**Major remarks**

**p 7, eq. (2)** Regarding the robustness of your method: you calculate the residual of the SLP *trend*. What happens if you exchange the order of operations, i.e., calculate the trend of SLP *residuals* (remove the first five SLP PCs from the SL fields and than look at the trends)?

**p 8, 1st para** so only 10-20% of the decadal SL trends are not directly related to SLP. How much is that in cm - I mean, what are we talking about?

**p 8, l 21/22** It is not clear from the caption (nor from the text!) what you are doing. As far as I understand the top row is correlation between full decadal SLP trend and full SL trend. In the bottom row the SLP signal is removed from the tide gauges, but what about SLP? Do you correlate the decadal SLP trend with the tide gauge residuals, or do you also subtract the first five PCs from the SLP trends? - Note that this remark not only applies to this figure, but to the whole paper.

**p 9, Tab. 1** What do you mean by "previous season"? You show four correlations. Take for instance "winter". You correlate it with winter - which winter? The same, or the following? You also correlate it with summer, but winter is not previous to summer. I guess that what you are doing is a lag-correlation with lag of 0, 1, 2, and 3 seasons, but I am not sure. Please clarify.

**p 8, l 27/28** Why would you expect a relation between air temp and SL in winter? I could imagine a relation between summer T and autumn (or winter) SL because of evaporation, but why winter? Which brings me to another question: Why do you consider precip in the following, but not evaporation?

**p 9, Fig. 8/9** precip is probably not independent of SLP. So if you remove the effect of SLP from your analysis, you probably also remove a lot of the precip effect. I guess that the purpose of these two figures is to somehow disentangle the two effects, but I do not understand what the result is. Does precip have an effect beyond SLP?

**p 10, l 3-11** Are you saying that for some (but not all) seasons precip affects SL in the following season, and that it depends on data set (reanalysis *vs.* CRU) for which seasons you find an effect? OK, so what, what are the implications?

**Detailed comments**

**p 1, l 23/24** Sounds odd - "decadal trend" depending on previous season precip. What you mean is that previous season precip also ha a decadal trend.

**p 3, l 8/9** To my knowledge the physical connection between the North Sea and the Baltic is through the Danish Straits, which are more or less exactly north-south (i.e., meridionally) oriented.

**p 3, l 13** Of course is the impact of NAO higher in winter than in summer. NAO is mainly a winter phenomenon. The explained variance of a NAO-like pattern is highest in winter, and small in summer.

**p 4, l 6** remove GIA effect → remove the GIA effect

**p 4, l 6** I would start a new paragraph after "time series"

**p 4, l13-15** too long a sentence and not to follow.

**p 4, l 18** of Atlantic Multidecadal Oscillation → of the Atlantic Multidecadal Oscillation

**p 6, l 3** continues → continuing

**p 7, l 1** the $\beta$ coefficients in this equations are different from those in eq. (1). Please use different symbols to prevent confusion.

**p 7, l 2** SLP principal component - I think yo mean the PC of SLP-*trend*, don't you?

**p 7, l 19** NAO is major factor → NAO is the major factor

**p 7, l 21** as it stands, this sentence implies that Stockholm is representative for the southern Baltic and Warnemünde for the Baltic proper.

**p 8, l 21/22 & l 29/30** The lower row is not explained.
**p 9, l 11**  tide gauge residuals → tide gauge trend residuals ????

**p 9, l 28**  delete second appearance of *between*

**Figures**  (i) Please add an indication of significance to all correlation maps. (ii) Consider removing panels from the figures. Having correlations for different seasons or for the two stations does not add significant information.

---

## Referee Comment (RC2) · Anonymous Referee #2 · 18 Apr 2017

In the present study "Mechanisms of variability of decadal sea-level trends in the Baltic Sea over the 20th century" the authors use long tide gauge records and reconstructions of different climatic variables to study large-scale factors influencing trends in the Baltic Sea level. Regional sea level trends can deviate strongly from global trends and therefore it is of great importance to understand the factors influencing sea level trends at regional scales. Thus, the present study could give valuable new insights into the factors influencing regional sea-level trends in the Baltic Sea. However, I have some concerns regarding this manuscript and I would recommend a major revision before the study could be published in Earth System Dynamics. I will list my concerns and comments below.

Major comments:

[Figure]

A) The authors present an interesting approach by filtering the direct influence of the atmospheric forcing on the sea level trends and only looking at the residuals. However, they do not show how relevant these residuals are. On page 8, 1st paragraph they only mention that for their regression model they use the first 5 principal components of the SLP trends that explain around 80% of the variance of SLP trends. But how large are the residuals of the regression analysis for the sea level trends? And how much of the variance of the sea level trends do these residuals explain?

B) The data sets used all cover different time periods. From the figures and the text it is not always clear which time period is used for which analysis. For consistency it would be best to use the common time period from 1901-2012 for all analysis except for the SSHA reconstructions and the NCEP/NCAR precipitation reconstruction, where it should be clearly indicated that only shorter time series are used. Further, I am missing a discussion of the quality of the data sets and possible problems with the data sets especially during the first decades.

C) A lot of the analyses are based on correlations, which in some cases are quite small. However, it is not shown if these correlations are significant. I would suggest to only plot the significant correlations in shading and the rest just as contours. (See also my comments on the figures below.)

D) The conclusion section is quite short and I am missing a discussion of the results and their implications.

E) The presentation of the figures should be improved. (See below for detailed comments.)

Further comments:

1.) The abstract should be rewritten to be more concise. For example, most of the 2nd paragraph could be cut and instead a stronger focus should be on the results.

2.) Page 8, line 11 and Figure 5: Why are only 9 tide gauges considered and not the

full 29?

3.) Page 8, line 31: The results are not very surprising since this was the aim of the approach, but the figures do not really add any new information. Therefore, I would only put them in the supplementary material.

4.) Figures 3, 4, 5, 6, 7, 8, and 9: The colour bar is not very well chosen. It is difficult to distinguish the colours for correlations between 0.6 and 1.0 and -0.5 and -1.0. I would suggest to only plot the significant correlations in colour and otherwise just the contours for example. And then to use a better separated colour scheme for the higher correlations. Further, in the multi-panel figures I would only plot one colour bar next to the whole figure and not individually for each panel. Instead I would make the subfigures larger.

5.) Figures 4, 5, 8, 9, 10: The positioning of the subfigures is a bit confusing. I would suggest to put Stockholm in the left column and Warnemünde in the right column and then arrange by season from top to bottom.

6.) Figure 4: I would crop the figures to focus on the Baltic Sea region since the correlations over the Atlantic are not discussed anyway.

7.) Figure 10: The titles are way too small and the colours are not explained.

---

## Referee Comment (RC3) · Anonymous Referee #3 · 8 May 2017

The paper is devoted to decadal trends of sea-level in the Baltic Sea over the twentieth century. Due to global climate change the question of sea level trends and their reasons is vital also for the Baltic Sea. Former studies about the inter-annual to decadal variations of the sea-level of the Baltic Sea have shown that an important part of variability in these time scales results from atmospheric forcing, that could be described through NAO index, mostly in winter. But it has been shown also that in summer the influence of precipitation and temperature has a strong effect on sea-level variations. In this article, a statistical model is used to capture the simultaneous link between atmospheric circulation and sea-level for seasonal means, and later residuals of statistical model are researched to reveal other reasons of sea-level variability. The paper is in many places unclear and I think that several points should be better addressed before considering it for a final publication.

[Figure]

1) The novelty of the paper is not clearly stated. 2) It is not clearly written out what is the consideration of using 11-year gliding trends. Why to correlate the speeds of change of various climatic variables? 3) There is no discussion part and the conclusions are very general. 4) The overall presentation is well structured, but in the section 2, there are too many subtitles, not every dataset needs a subtitle. 5) The methodology section needs improvements. It is not clearly written what is the period under consideration, various data sets have various periods of availability. NAO indices are available from different sites. There is no reference to dataset used. Were the gridded data used in the original grid, or were computed into a common grid? What is the study area? Page 7 line 3 "Y is time series of gliding trend anomaly". Anomaly against what? It is not explained.

Specific comments:

1) Typing errors in references 2) The quality of the figures should be improved to add readability to this work. In Fig 1 the numbers of stations are partly covered by the colour point. Would be better to present the names of the stations in Fig 2, then is seen easily how long are the time series in separate stations. Knowing of names of stations in Fig 1 is not crucial. In figure caption of Fig 10 the blue and red line are not explained. What are units of trend?

---

## Author Comment (AC2) · 2 Jun 2017

*In the present study "Mechanisms of variability of decadal sea-level trends in the Baltic Sea over the 20th century" the authors use long tide gauge records and reconstructions of different climatic variables to study large-scale factors influencing trends in the Baltic Sea level. Regional sea level trends can deviate strongly from global trends and therefore it is of great importance to understand the factors influencing sea level trends at regional scales. Thus, the present study could give valuable new insights into the factors influencing regional sea-level trends in the Baltic Sea. However, I have some concerns regarding this manuscript and I would recommend a major revision before the study could be published in Earth System Dynamics. I will list my concerns and comments below.*

*Major comments:*

*A) The authors present an interesting approach by filtering the direct influence of the atmospheric forcing on the sea level trends and only looking at the residuals. However, they do not show how relevant these residuals are. On page 8, 1st paragraph they only mention that for their regression model they use the first 5 principal components of the SLP trends that explain around 80% of the variance of SLP trends. But how large are the residuals of the regression analysis for the sea level trends? And how much of the variance of the sea level trends do these residuals explain?*

We will clarify the text and comment on decadal SL trend variations, decadal SL residual trend variations.

In the following figure, we show the decadal running trends of observations (blue) and of residuals (green) for the Stockholm station in wintertime.

[Figure]

Here, we consider the Stockholm station for the wintertime over the period 1900-2012. The sea-level trend residuals explain 41% variance of sea-level trends. The maximum (minimum) value of sea-level trend residuals in this period is 9.5 (-21.1) mm/year.

*B) The data sets used all cover different time periods. From the figures and the text it is not always clear which time period is used for which analysis. For consistency it would be best to use the common time period from 1901-2012 for all analysis except for the SSHA reconstructions and the NCEP/NCAR precipitation reconstruction, where it should be clearly indicated that only shorter time series are used. Further, I am missing a discussion of the quality of the data sets and possible problems with the data sets especially during the first decades.*

We indeed always use the same period of analysis.

For the analysis, we used PSMSL (www.psmsl.org) data sets. These data sets are quality controlled.

*C) A lot of the analyses are based on correlations, which in some cases are quite small. However, it is not shown if these correlations are significant. I would suggest to only plot the significant correlations in shading and the rest just as contours. (See also my comments on the figures below.)*

We will include a contour indicating the significant correlations.

*D) The conclusion section is quite short and I am missing a discussion of the results and their implications.*

We will expand the conclusion and focus more on implications of the results.

*E) The presentation of the figures should be improved. (See below for detailed comments.)*

We will improve the quality of Figure presentations.

*Further comments:*

*1.)The abstract should be rewritten to be more concise. For example, most of the 2$^{nd}$ paragraph could be cut and instead a stronger focus should be on the results.*

We will rewrite the abstract according to this suggestion.

*2.) Page 8, line 11 and Figure 5: Why are only 9 tide gauges considered and not the full 29?*

We consider these tide gauges to be representative of the Baltic Sea, well distributed over the Baltic Sea region and have long records.

*3.) Page 8, line 31: The results are not very surprising since this was the aim of the approach, but the figures do not really add any new information. Therefore, I would only put them in the supplementary material.*

We will move them to the supplementary material.

*4.) Figures 3, 4, 5, 6, 7, 8, and 9: The colour bar is not very well chosen. It is difficult to distinguish the colours for correlations between 0.6 and 1.0 and -0.5 and -1.0. I would suggest to only plot the significant correlations in colour and otherwise just the contours for example. And then to use a better separated colour scheme for the higher correlations. Further, in the multi-panel figures I would only plot one colour bar next to the whole figure and not individually for each panel. Instead I would make the subfigures larger.*

We will improve the quality of correlation scale based on the suggestion. We will plot only one colour bar next to the whole figure and expand the subfigures.

*5.) Figures 4, 5, 8, 9, 10: The positioning of the subfigures is a bit confusing. I would suggest to put Stockholm in the left column and Warnemünde in the right column and then arrange by season from top to bottom.*

We will reorder the positions of the subfigures based on the suggestion.

*6.) Figure 4: I would crop the figures to focus on the Baltic Sea region since the correlations over the Atlantic are not discussed anyway.*

We will modify the representation of the figures accordingly.

*7.) Figure 10: The titles are way too small and the colours are not explained.*

We will add explanation of colours and make the size of the titles larger.

---

## Author Response (AR1)

[revised manuscript text omitted]

We thank the reviewer very much for reviewing our manuscript, for providing constructive criticism and useful suggestions. We respond to all comments below.
*The paper analyses the sea level (SL) variability in the Baltic Sea and its drivers. Sea level observations from 29 tide gauges, some of them going back to the eighteenth century, from around the Baltic Sea are used together with a satellite-based reconstruction of sea level for the whole Baltic that goes back to 1950. Observation-based data sets of SLP, precip, and temperature are used to investigate their relation to SL variations. The paper focuses on longer time scales by considering the relation between decadal-scale trends of the variables, rather than the variables themselves.*

*SL variations on longer time scales are found to be highly connected to NAO (North Atlantic Oscillation) variations, with larger correlations in the northern than in the southern Baltic. Precipitation in the Baltic catchment also plays an important role for SL variability.*

*Discussion*
*The paper seems to be technically sound, but I miss an explanation of the relevance of the results. What are possible implications? Furthermore, the presentation is often unclear. The list below gives some hints as to where the presentation of the work can be improved. Together, I think that a major re-writing of the paper is necessary before it can be accepted.*

Previous studies have shown that the sea-level records display relatively large variations of decadal trends in the Baltic Sea. This indicates that natural factors can cause substantial deviations from the expected spatially homogeneous centennial sea-level trend due to large-scale factors like rising ocean temperatures in the North Atlantic, melting of polar ice caps. These regional natural factors should be understood and taken into account, especially for shorter term (multidecadal) future sea-level projections. Whereas the mechanisms responsible for interannual variability have been more profusely studied, it is still not known whether the mechanisms that have been claimed to account for the interannual variations of sea-level are also responsible for the variability of decadal sea-level trends in the Baltic Sea.

In this study, we analyse long-term sea-level and climate records with the aim of explaining the observed variability of the decadal and multidecadal sea-level trends in the Baltic Sea. We mainly investigate whether the same mechanisms that have been found to explain the interannual variations

of Baltic sea-level are also responsible for the variability of the decadal sea-level trends.

We have clarified the novelty of the study in the manuscript.

*Major remarks*
*p 7, eq. (2) Regarding the robustness of your method: you calculate the residual of the SLP trend. What happens if you exchange the order of operations, i.e., calculate the trend of SLP residuals (remove the first five SLP PCs from the SL fields and than look at the trends)?*

We made a multivariate regression analysis between first 5 principal vectors of SLP fields and sea-level for the period 1900-2013 on the interannual time scale. The analysis suggested by the reviewer is compared in the following figure.

[Figure]

In the Figure, the 11-year gliding trend residuals of multivariate regressions between Stockholm sea-level and first 5 PCs of SLP field on the interannual sea-level variability(blue) and between Stockholm sea-level and first 5 PCs of SLP field based on 11-year gliding trend time series(green) are shown for wintertime. The correlation between residuals is 0.92. This confirms the robustness of our results.

*p 8, 1st para so only 10-20% of the decadal SL trends are not directly related to SLP. How much is that in cm - I mean, what are we talking about?*

These explained variances only show how much variance of the SLP field trends can be explained by first five principal components of the SLP trends, and not the amount of SL variance that is explained by the SLP field. We have written that part to clarify that those explained variances driven from SLP PCA and not related to the SL gliding trends.

*p 8, l 21/22 It is not clear from the caption (nor from the text!) what you are doing. As far as I understand the top row is correlation between full decadal SLP trend and full SL trend. In the bottom row the SLP signal is removed from the tide gauges, but what about SLP? Do you correlate the decadal SLP trend with the tide gauge residuals, or do you also subtract the first five PCs from the SLP trends? – Note that this remark not only applies to this figure, but to the whole paper.*

The figures display the correlation of the SL residuals with the complete SLP field. They are intended to show that the residuals do not really contain any simultaneous SLP signal.

We have rewritten and clarified that caption.

*p 9, Tab. 1 What do you mean by "previous season"? You show four correlations. Take for instance "winter". You correlate it with winter - which winter? The same, or the following? You also correlate it with summer, but winter is not previous to summer. I guess that what you are doing is a lag-correlation with lag of 0, 1, 2, and 3 seasons, but I am not sure. Please clarify.*

The reviewer is right. We have updated the table and used lag0, lag1, lag2, lag3 notation.

*p 8, l 27/28 Why would you expect a relation between air temp and SL in winter? I could imagine a relation between summer T and autumn (or winter) SL because of evaporation, but why winter? Which brings me to another question: Why do you consider precip in the following, but not evaporation?*

We explore here the indirect correlation between SL and temperature in wintertime, mediated by the atmospheric circulation. The NAO is correlated with both SL and air temperature. We were not referring to a causal relationship between winter temperature and SL. This paragraph will be reformulated.

We have done calculation with considering the evaporation (P-E) and shown the results in Table 1. This paragraph is also partly reformulated.

*p 9, Fig. 8/9 precip is probably not independent of SLP. So if you remove the effect of SLP from your analysis, you probably also remove a lot of the precip effect. I guess that the purpose of these two figures is to somehow disentangle the two effects, but I do not understand what the result is. Does precip have an effect beyond SLP?*

The reviewer is right. We cannot disentangle the effect of precipitation in one season from the effect of SLP in the same season, and some of the effects of precipitation will also be filtered out when considering the SL residuals. However, the effect - in any - of precipitation in the previous season (lag -1) should still be contained in the SL residuals. This is the effect we are looking for.

*p 10, l 3-11 Are you saying that for some (but not all) seasons precip affects SL in the following season, and that it depends on data set (reanalysis vs. CRU) for which seasons you find an effect? OK, so what, what are the implications?*

CRU data is available only over land, whereas reanalysis data, though imperfect, also over the

whole basin. We think that it may be the main reason for the different results.

***Detailed comments***
*p 1, l 23/24 Sounds odd - "decadal trend" depending on previous season precip. What you mean is that previous season precip also has a decadal trend.*

We have clarified the text.

*p 3, l 8/9 To my knowledge the physical connection between the North Sea and the Baltic is through the Danish Straits, which are more or less exactly north-south (i.e., meridionally) oriented.*

Old version: "*its narrow physical connection to the North Sea and North Atlantic is also zonally oriented*"

New version: "*it is connected to the North Sea by narrow straits.*"

*p 3, l 13 Of course is the impact of NAO higher in winter than in summer. NAO is mainly a winter phenomenon. The explained variance of a NAO-like pattern is highest in winter, and small in summer.*

We agree with the reviewer's comment, but we are unsure as to how it prompts us to change the text.

*p 4, l 6 remove GIA effect ! remove the GIA effect*

We have changed this accordingly.

*p 4, l 6 I would start a new paragraph after "time series"*

We have changed this accordingly.

*p 4, l13-15 too long a sentence and not to follow.*

We have clarified the text.

*p 4, l 18 of Atlantic Multidecadal Oscillation ! of the Atlantic Multidecadal Oscillation*

We have changed this accordingly.

*p 6, l 3 continues ! continuing*

We have changed this accordingly.

*p 7, l 1 the _ coefficients in this equations are different from those in eq. (1). Please use different symbols to prevent confusion.*

We have changed this accordingly.

*p 7, l 2 SLP principal component - I think yo mean the PC of SLP-trend, don't you?*

We have changed this accordingly.

*p 7, l 19 NAO is major factor ! NAO is the major factor*

We have changed this accordingly.

*p 7, l 21 as it stands, this sentence implies that Stockholm is representative for the southern Baltic and Warnemünde for the Baltic proper.*

We have changed this accordingly.

*p 8, l 21/22 & l 29/30 The lower row is not explained.*

We have added the explanation.

*p 9, l 11 tide gauge residuals ! tide gauge trend residuals ????*

We have changed this accordingly.

*p 9, l 28 delete second appearance of between*

We have changed this accordingly.

*Figures*
   *(i)  Please add an indication of significance to all correlation maps.*

   We have added the significance information for all correlation maps.

   (ii)  *Consider removing panels from the figures. Having correlations for different seasons or for the two stations does not add significant information.*

   We have removed unneeded panels from the figures.

We thank the reviewer very much for reviewing our manuscript, for providing constructive criticism and useful suggestions. We respond to all comments below.
*In the present study "Mechanisms of variability of decadal sea-level trends in the Baltic Sea over the 20th century" the authors use long tide gauge records and reconstructions of different climatic variables to study large-scale factors influencing trends in the Baltic Sea level. Regional sea level trends can deviate strongly from global trends and therefore it is of great importance to understand the factors influencing sea level trends at regional scales. Thus, the present study could give valuable new insights into the factors influencing regional sea-level trends in the Baltic Sea. However, I have some concerns regarding this manuscript and I would recommend a major revision before the study could be published in Earth System Dynamics. I will list my concerns and comments below.*

*Major comments:*

*A) The authors present an interesting approach by filtering the direct influence of the atmospheric forcing on the sea level trends and only looking at the residuals. However, they do not show how relevant these residuals are. On page 8, 1st paragraph they only mention that for their regression model they use the first 5 principal components of the SLP trends that explain around 80% of the variance of SLP trends. But how large are the residuals of the regression analysis for the sea level trends? And how much of the variance of the sea level trends do these residuals explain?*

In the following figure, we show the decadal running trends of observations (blue) and of residuals (green) for the Stockholm station in wintertime.

[Figure]

Here, we consider the Stockholm station for the wintertime over the period 1900-2012. The sealevel trend residuals explain 41% variance of sea-level trends. The maximum (minimum) value of sea-level trend residuals in this period is 9.5 (-21.1) mm/year.

We have clarified the text and commented on decadal SL trend variations, decadal SL residual trend variations.

*B) The data sets used all cover different time periods. From the figures and the text it is not always clear which time period is used for which analysis. For consistency it would be best to use the common time period from 1901-2012 for all analysis except for the SSHA reconstructions and the NCEP/NCAR precipitation reconstruction, where it should be clearly indicated that only shorter time series are used. Further, I am missing a discussion of the quality of the data sets and possible problems with the data sets especially during the first decades.*

We indeed always use the same period of analysis.

For the analysis, we used PSMSL (www.psmsl.org) data sets. These data sets are quality controlled.

*C) A lot of the analyses are based on correlations, which in some cases are quite small. However, it is not shown if these correlations are significant. I would suggest to only plot the significant correlations in shading and the rest just as contours. (See also my comments on the figures below.)*

We have included a contour indicating the significant correlations.

*D) The conclusion section is quite short and I am missing a discussion of the results and their implications.*

We have expanded the conclusion and focused more on implications of the results.

*E) The presentation of the figures should be improved. (See below for detailed comments.)*

We have tried to improve the quality of Figure presentations.

*Further comments:*

*1.)The abstract should be rewritten to be more concise. For example, most of the 2$^{nd}$ paragraph could be cut and instead a stronger focus should be on the results.*

We have rewritten the abstract according to this suggestion.

*2.) Page 8, line 11 and Figure 5: Why are only 9 tide gauges considered and not the full 29?*

We consider these tide gauges to be representative of the Baltic Sea, well distributed over the Baltic Sea region and have long records.

*3.) Page 8, line 31: The results are not very surprising since this was the aim of the approach, but the figures do not really add any new information. Therefore, I would only put them in the supplementary material.*

In the first round, we mentioned that we will remove related part to the supplementary material. However, another reviewer has found this part important and wanted to see further explanation on

that part, thus, we have decided to keep this part in the manuscript.

*4.) Figures 3, 4, 5, 6, 7, 8, and 9: The colour bar is not very well chosen. It is difficult to distinguish the colours for correlations between 0.6 and 1.0 and -0.5 and -1.0. I would suggest to only plot the significant correlations in colour and otherwise just the contours for example. And then to use a better separated colour scheme for the higher correlations. Further, in the multi-panel figures I would only plot one colour bar next to the whole figure and not individually for each panel. Instead I would make the subfigures larger.*

We have tried to improve the quality of Figures scale based on the suggestion

*5.) Figures 4, 5, 8, 9, 10: The positioning of the subfigures is a bit confusing. I would suggest to put Stockholm in the left column and Warnemünde in the right column and then arrange by season from top to bottom.*

We have reordered the positions of the subfigures based on the suggestion.

*6.) Figure 4: I would crop the figures to focus on the Baltic Sea region since the correlations over the Atlantic are not discussed anyway.*

We have modified the representation of the figures accordingly.

*7.) Figure 10: The titles are way too small and the colours are not explained.*

We have updated Figure, added explanation of colours and made the size of the titles larger.

We thank the reviewer very much for reviewing our manuscript, for providing constructive criticism and useful suggestions. We respond to all comments below.
*The paper is devoted to decadal trends of sea-level in the Baltic Sea over the twentieth century. Due to global climate change the question of sea level trends and their reasons is vital also for the Baltic Sea. Former studies about the inter-annual to decadal variations of the sea-level of the Baltic Sea have shown that an important part of variability in these time scales results from atmospheric forcing, that could be described through NAO index, mostly in winter. But it has been shown also that in summer the influence of precipitation and temperature has a strong effect on sea-level variations. In this article, a statistical model is used to capture the simultaneous link between atmospheric circulation and sea-level for seasonal means, and later residuals of statistical model are researched to reveal other reasons of sea-level variability. The paper is in many places unclear and I think that several points should be better addressed before considering it for a final publication.*

*1) The novelty of the paper is not clearly stated.*

Previous studies have shown that the sea-level records display relatively large variations of decadal trends in the Baltic Sea. This indicates that natural factors can cause substantial deviations from the expected spatially homogeneous centennial sea-level trend due to large-scale factors like rising ocean temperatures in the North Atlantic, melting of polar ice caps. These regional natural factors should be understood and taken into account, especially for shorter term (multidecadal) future sea-level projections. Whereas the factors that drive the interannual variations of Baltic sea level have been more profusely investigated, it is still not known whether the mechanisms that have been claimed to account for the interannual variations of sea-level are also responsible for the variability of decadal sea-level trends in the Baltic Sea.

In this study, we analyse long-term sea-level and climate records with the aim of explaining the observed variability of the decadal and multidecadal sea-level trends in the Baltic Sea. We mainly investigate whether the same mechanisms that have been found to explain the

interannual variations of Baltic sea-level are also responsible for the variability of the decadal sea-level trends.

We have clarified the novelty of the study in the manuscript.

*2) It is not clearly written out what is the consideration of using 11-year gliding trends. Why to correlate the speeds of change of various climatic variables?*

The increasing external climate forcing impacts the global mean temperature, so that in increasing forcing results in higher temperatures. In contrast to temperature, the effect of the increasing radiative forcing is, to first approximation, related to the sea-level rate. The sea-level itself, in contrast to the sea-level rate, reflects the cumulative impact of past external climate forcing. This is why the studies focused on the detection and attribution of climate change deal with sea-level rates and their variability. There is therefore a need to characterise the mechanisms that may affect the variations of the sea-level rate. In the Baltic Sea, previous studies have investigated the link between climate or atmospheric forcing and sea-level. However, so far very few studies have focused on the sea-level rates and on the question of whether the mechanism that affect sea-level variability are also as important for the decadal sea-level rates or whether other mechanisms come into play.

*3) There is no discussion part and the conclusions are very general.*

We have expanded the conclusion and focused more on implications of the results.

We have included a discussion section addressing several points: the magnitude of atmosphere-driven decadal trends versus the residuals trends, the differences between the mechanism that we have identified as driving factors for decadal sea-level variability and the factors that are responsible for the interannual variations, and in particular the possible role of precipitation.

The conclusion section is shortened and tightened up summarizing the most import points that can be derived from the results: the variability of the decadal trends in the Baltic is spatially more homogeneous than the interannual variations; the factors that are responsible are regional and not clearly connected to the North Atlantic; trends in wind forcing can only explain about 50% of the trend variability; precipitation may play a relevant role.

*4) The overall presentation is well structured, but in the section 2, there are too many subtitles, not every dataset needs a subtitle.*

We have updated the text accordingly.

*5) The methodology section needs improvements. It is not clearly written what is the period under consideration, various data sets have various periods of availability. NAO indices are available from different sites. There is no reference to dataset used. Were the gridded data used in the original grid, or were computed into a common grid? What is the study area? Page 7 line 3 "Y is time series of gliding trend anomaly". Anomaly against what? It is not explained.*

We have clarified the methodology section based on the reviewer's suggestions.

*Specific comments:*

*1) Typing errors in references*

We have corrected them.

*2) The quality of the figures should be improved to add readability to this work. In Fig 1 the numbers of stations are partly covered by the colour point. Would be better to present the names of the stations in Fig 2, then is seen easily how long are the time series in separate stations.*

We have added the names of the stations in Figure 2.

*Knowing of names of stations in Fig 1 is not crucial. In figure caption of Fig 10 the blue and red line are not explained*

We have replotted the Figure and explained the colours.

*What are units of trend?*

Decadal gliding trends that are shown in Figure 10 are unitless, since those decadal trends of sea level and AMO-index are standardized through dividing gliding decadal trend time series by their standard deviations.

---

## Author Response (AR2)

[revised manuscript text omitted]

We thank reviewers for their further help with the manuscript. All suggestions of the reviewers are considered and accordingly carried out in the manuscript. Please see our point-by-point responses below.

**REVIEWER #1**

*I am satisfied with how the authors dealt with my comments.*

*I spotted a few typos:*

*p 4, l 11: delete "causes"*

**Answer:** The word "causes" has been deleted.

*p 7, l 10: linkage -> link*

**Answer:** The word has been accordingly updated.

*p 7, l 32: delete "show"*

**Answer:** The word "show" has been deleted.

*caption Fig. 5: what is shown in the left columns? (only the right columns are mentioned)*

Mentioned point has been clarified in the caption.

*REVIEWER #2*

*In the present revised version of the paper "Mechanisms of variability of decadal sea-level trends in the Baltic Sea over the 20th century" the authors have addressed most of the reviewers comments. However, I still think that some major revisions are necessary in order for the manuscript to be publishable in ESD.*

*i) The authors responded to my major comment A) in the response letter by showing a comparison of the sea level trends and the sea level trend residuals. However, they did not include it in the manuscript. I still think that it is important for the reader to know how much of the sea level trends can be explained by the residuals and is thus independent of the direct atmospheric forcing. Further, the sentence in line 32-33, page 7 makes no sense. Maybe remove the "show"?*

**Answer:** We had added a comparison of corresponding results through the first revision session. It can be found on Page 9, Line 14-19 in the first revised manuscript.

The word "show" has been deleted.

*ii) Again, the authors responded to my major comment B) in the response letter, but in the manuscript it is still not always clear, what time period is used for the analyses.*

**Answer:** We had previously added a sentence in order to clarify the analysis period during the first revision phase. It can be found in the Methodology section (Page 6 Line 9-10) of the first revised manuscript.

*iii) The authors now included contours or added a sentence to indicate the significance of the shown correlations. But what method was used to calculate the significance levels?*

**Answer:** For any record length (which describes the degree of freedom), any value of significance is computed at the 95% confidence level, under the usual assumptions of normally distributed and temporally uncorrelated variables.

The following sentence has been added to the last part of the Methodology section:

*"A correlation is considered significant when it passes the 95% level under the usual assumptions of normally distributed and temporally uncorrelated variables. For this estimation we used the usual t-distribution test"*

*iv) The conclusion section is still very short and the implications of the results are not sufficiently discussed.*

**Answer:** This section has been expanded with some implications of our study for future work.

*v) Again, the authors reply to my comment 2.) on the number of the tide gauges shown only in the response letter, but the information should also be included in the manuscript.*

**Answer:** We specially thank reviewer for this suggestion.

The following sentence has been added to the manuscript (above the Figure 5 caption):

*"We consider these tide gauges, nine tide gauges in total, to be representative of the Baltic Sea, reasonable span the Baltic Sea region and have long records."*

*vi) The labelling of the figures needs to be improved. It should be clear from the figures, which variable is show at what location in which season. Especially, in multi-panel figures like Fig. 5 it is very confusing otherwise.*

**Answer:** For this recommendation, we have now added labels including the information of variable and seasons to the Figures 4, 5, 6, 7, 8 and 9. We have also improved the captions of the Figures 4, 5, 6 and 7. We think that adding more labels to the single figures would impair the view of figures.

---

## Author Response (AR3)

Dear authors,

I am pleased to inform you that your manuscript is now accepted. Please perform the following minor technical corrections:

**Answer:** We are glad that our manuscript has been accepted to be published in the special issue *"Multiple drivers for Earth system changes in the Baltic Sea region"* of Earth System Dynamics. We are deeply grateful to the Editor for attentively guiding whole process and for his very useful suggestions which helped us substantially improve the quality of the manuscript. Please, find our point-by-point responses to the technical corrections below.

1) to my knowledge two hyphens per word are not used. I suggest on P3, L14 sea level pressure and P5, L20 sea surface temperature

**Answer:** The text has been updated accordingly.

2) please check all references, e.g. Enfield et al. 2001

**Answer:** We specially thank the Editor for this careful feedback. We now used proper "Copernicus Template" for references and the reference lists and in the in-text citations.

3) please explain the colors used in Fig.2 in the figure caption (I assume there is no meaning?)

**Answer:** As Editor assumed, there is no meaning of the colours. The Figure was previously updated and improved because of one of the referees' suggestion. They are coloured to separate time series over the lines.

This sentence has been added to the caption: "*Time series are coloured for better visibility.*"

4) Figure 8, caption: ...area averaged CRU decadal trends in precipitation ...

**Answer:** The text has been updated accordingly.

[revised manuscript text omitted]